

# Calculating the turbulent fluxes in the atmospheric surface layer with neural networks

Lukas Hubert Leufen[1,2] and Gerd Schädler[1]

[1]Institute of Meteorology and Climate Research - Department Troposphere Research, Karlsruhe Institute of Technology, Karlsruhe, Germany
[2]now at Forschungszentrum Jülich GmbH, Jülich, Germany

**Correspondence:** G. Schädler (gerd.schaedler@kit.edu)

**Abstract.** The turbulent fluxes of momentum, heat and water vapour link the Earth's surface with the atmosphere. The correct modelling of the flux interactions between these two systems with very different time scales is therefore vital for climate (resp. Earth system) models. Conventionally, these fluxes are modelled using Monin-Obukhov similarity theory (MOST) with stability functions derived from a small number of field experiments; this results in a range of formulations of these functions and thus also in the flux calculations; furthermore, the underlying equations are non-linear and have to be solved iteratively at each time step of the model. For these reasons, we tried here a different approach, namely using an artificial neural network (ANN) to calculate the fluxes resp. the scaling quantities $u_*$ and $\theta_*$, thus avoiding explicit formulas for the stability functions. The network was trained and validated with multi-year datasets from seven grassland, forest and wetland sites worldwide using the Broyden–Fletcher–Goldfarb–Shanno (BFGS) quasi-Newton backpropagation algorithm and six-fold cross validation. Extensive sensitivity tests showed that an ANN with six input variables and one hidden layer gave results comparable to (and in some cases even slightly better than) the standard method. Similar satisfying results were obtained when the ANN routine was implemented in a one-dimensional stand alone land surface model (LSM), opening the way to implementation in three-dimensional climate models. In case of the one-dimensional LSM, no CPU time was saved when using the ANN version, since the small time step of the standard version required only one iteration in most cases. This could be different in models with longer time steps, e.g. global climate models.

## 1 Introduction

The turbulent fluxes of momentum, heat, water vapour and trace gases link the atmosphere with the Earth's surface. The faithful representation of these fluxes is therefore essential for a proper functioning of climate models (and also for weather forecast models). In these models, the fluxes respective covariances are parameterised using a velocity scale $u_*$ and a (potential) temperature scale $\theta_*$ as momentum flux $\tau = \rho u_*^2$ and heat flux $H = -\rho c_p u_* \theta_*$ ($\rho$ is air density, $c_p$ is air heat capacity). $u_*$ and $\theta_*$ depend on near surface wind and temperature, their gradients, surface roughness and atmospheric stability. In the framework of the almost exclusively used Monin-Obukhov similarity theory (MOST; Monin and Obukhov, 1954), one has to determine stability functions for momentum and heat which depend on a single stability parameter (for details, see e.g. Arya, 2001). These stability functions must be determined empirically and were obtained by different authors from regressions on observations





from a small number of field experiments. As shown in Högström (1996), the results vary considerably, especially in the very stable and the very unstable regimes, due to a lack resp. a large scatter of the observations and possibly violations of the assumptions of MOST. Furthermore, the underlying non-linear equations must be solved iteratively at each time step of a model run which can be time consuming.

In the present study, artificial neural networks (ANN) and their ability to simulate a wide range of relationships between input and output variables as a universal approximator (Hornik et al., 1989) are used to model the stability functions. Our ultimate goal is to replace the relevant subroutines in a climate model by ANNs in order to improve overall model performance. A good overview of various applications of ANNs in different disciplines can be found in Zhang (2008). Several studies (e.g. Gardner and Dorling, 1999; Elkamel et al., 2001; Kolehmainen et al., 2001) describe applications of ANNs to meteorological

and air quality problems. In these studies, long time series of data were available for ANN training and only one station was involved in the training and validation process. Comrie (1997) finds that neural networks are "somewhat, but not overwhelmingly" better than regression models. Best performance was obtained with an ANN incorporating lagged data. Gomez-Sanchis et al. (2006) use a multi-layer perceptron (MLP) to predict ozone concentrations near Valencia based on meteorological and traffic information. Different model architectures were tested and good agreement with observations was found. However, dif-

ferent years required different model architectures. Elkamel et al. (2001) used a one hidden layer ANN and meteorological and precursor concentrations to predict ozone levels in Kuwait. They found that the ANN gave consistently better predictions than linear and nonlinear regression models. Kolehmainen et al. (2001) compare the ability of self-organising maps and MLP to predict $NO_2$ concentrations when combined with different methods to preprocess the data. They find that direct application of the MLP gave best results. In all these studies just one hidden layer was sufficient and it was pointed out that careful selection

of input data was crucial. An interesting application of a neural network's ability to learn is Knutti et al. (2003): they teached a neural network to simulate certain output variables of a global climate model and used the result to establish probability density functions and to enlarge a global climate model ensemble considerably. Gentine et al. (2018) use an ANN to parameterise the effects of subgrid scale convection in a global climate model. The ANN learns the combined effects of turbulence, radiation and cloud microphysics from a convection resolving submodel. They find that the ANN can predict many of these processes

skillfully and reduce inherent variance.

This paper is structured as follows: in Chapter 2, we give a short overview over Monin-Obuhkov similarity theory and artificial neural networks, introduce cross-validation, present the data used (including important quality checks) and describe our strategy to find the best network. Thereafter, trained ANNs (which are in fact MLPs, but we will stick to the generic name ANN here) are validated and results are discussed (Chapter 3). In Chapter 4, the best performing ANN is implemented in a

one-dimensional land surface model (LSM) and results are compared with the ones of the standard version. A summary is given in Chapter 5.



## 2 Methods and data

### 2.1 Monin-Obukhov similarity theory (MOST)

The turbulent fluxes of momentum, heat, water vapor and trace gases between the Earth's (land and water) surface and the atmosphere are usually calculated on the basis of Monin-Obukhov similarity theory (MOST, Monin and Obukhov (1954)). We give here a very short survey over MOST, focussing on momentum and heat fluxes; details can be found in Arya (2001). Assuming homogeneous terrain, quasistationary (i.e fair weather) conditions and small terrain roughness, MOST postulates that

turbulence in the surface (also called Prandtl or constant flux) layer depends only on four quantities: the height above ground resp. canopy $z$, a velocity scale $u_*$, a temperature scale $\theta_*$ and a buoyancy term $g/\theta$, where $g$ is gravitational acceleration and $\theta$ denotes potential temperature. According to the Buckingham Pi-Theorem, these four quantities based on length, time and temperature can be combined to a single non-dimensional quantity $\zeta = z/L$, where $L = u_*^2\theta/(\kappa g\theta_*)$ is the Obukhov length and $\kappa \approx 0.40$ is the von Kármán constant ; other dimensionless quantities like dimensionless wind and temperature gradients

can be expressed as functions of $\zeta$. The Obukhov length $L$ measures the stratification of the surface layer: large (positive or negative) values (i.e. $\zeta \approx \pm 0$) indicate neutral stratification, positive values indicate stable stratification, negative values indicate unstable stratification. Since momentum flux is expressed as $\tau = \rho u_*^2$, and heat flux as $H = -\rho c_p u_* \theta_*$ ($\rho$ is air density, $c_p$ is air heat capacity), our goal is to determine $u_*$ and $\theta_*$ from known quantities, which are in our case modelled or observed wind and temperature gradients in the surface layer.

Non-dimensional wind shear $\phi_m$ and the non-dimensional gradient of the potential temperature $\phi_h$ (also called stability functions) can be written as

$$\phi_m(\zeta) = \frac{\kappa z}{u_*}\frac{\partial u}{\partial z}, \quad \phi_h(\zeta) = \frac{\kappa z}{\theta_*}\frac{\partial \theta}{\partial z} \tag{1}$$

respectively, where $u$ is the mean wind speed at height $z$. The "'universal"' functions $\phi_m$ and $\phi_h$ can be obtained from simultaneously measured values of the wind and temperature gradients and the momentum and heat fluxes (providing $u_*$ and $\theta_*$). Data

from field experiments have been used to derive these universal functions, notably the Kansas experiment in 1968 by Businger et al. (1971). Generally, the stability functions thus obtained have the form

$$\phi_{m,h}(\zeta) = (\alpha_{m,h} + \beta_{m,h}\zeta)^{\gamma_{m,h}}$$

with the coefficients depending on $\zeta > 0$ or $\zeta \leq 0$. An overview of these functions can be found in Högström (1988); it is shown there that there is considerable scatter in the data (especially under very stable and very unstable conditions) and, as

a result, also in the derived universal functions. In applications, differences are known rather than gradients. Integrating the functions (1) between a reference height $z_r$ and $z$ yields

$$\kappa(u(z) - u(z_r))/u_* = ln(z/z_r) - \Psi_m(z/L), \quad \kappa(\theta(z) - \theta(z_r))/\theta_* = ln(z/z_r) - \Psi_h(z/L) \tag{2}$$



where

$$\Psi_{m,h}(z/L) = \int\limits_{z_r/L}^{z/L} (1 - \phi_{m,h}(u))du/u$$

For the purpose of climate modelling, i.e. obtaining fluxes from simulated wind and temperature profiles, $u_*$ and $\theta_*$ need to be derived from wind resp. temperature data at two heights and eqns. (1) or (2). Since $\zeta$ itself depends on $u_*$ and $\theta_*$, this amounts to solving a system of two nonlinear equations; we will call this traditional method the MOST method.

By employing neural networks in the context of MOST, we pursue two goals: first, we want to avoid the somewhat inflexible and uncertain approach of obtaining the scaling quantities $u_*$ and $\theta_*$ from profiles via stability functions; instead we want a

neural network to learn this relationship, based on larger and newer datasets. Second, once a neural network has learned the relationships, the iterative calculation of the scaling quantities during a simulation can be replaced by such a trained network, possibly saving computing time.

## 2.2   Neural networks

We describe here only those aspects of neural networks which are relevant to our study; for more information on neural

networks, the reader is referred the literature, e.g. (Rojas, 2013; Kruse et al., 2016). Neural networks, or more precisely artificial neural networks (ANN), are a widely used technique to solve classification and regression problems as well as to analyse time series (Zhang, 2008). The building blocks of an ANN are the so-called neurons which are connected within a network with each by weights other via input and output nodes. A neuron processes input data as follows:

$$o_j = f\left(\sum_i^N o_{i_k} \cdot w_{i_k j}\right),$$

where $o_j$ is the output of the neuron $j$, $N$ is the number of inputs, $o_{i_k}$ is the $i_k$th input and $w_{i_k j}$ is corresponding weight. A network becomes non-linear by using non-linear activation functions $f$. Each neuron belongs to a unique layer in a directed graph. Here, we use so-called multi-layer perceptrons (MLP), also known as feed-forward networks due to the unidirectional information flow. Each MLP consists of an input, an output and at least one hidden layer with an arbitrary number of neurons. The input layer takes (normalised) input data and the output data returns the (also normalised) results of the MLP. Normalisation

is essential for equal weighting of the input and for consistency with the activation functions. Input information is propagated from layer to layer while each neuron responds to the signal. Bias neurons are used to adjust the activation level.

All free parameters (i.e. weights) of a MLP need to be determined by a training process. In the case of supervised learning, the MLP knows its deviation from target values at every time and an error can be calculated using this deviation (Zhang, 2008). The aim of the training is to minimise an error metric by adjusting the network's weights. Here we use the mean squared error

(MSE)

$$MSE = \frac{1}{|P|} \sum_{p \in P} \frac{1}{N^\Omega} \sum_j^{N^\Omega} \left(t_{j,p} - o_{j,p}\right)^2 \qquad (3)$$

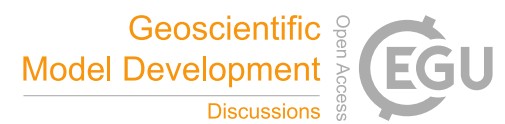



$P$ is the total number of data points, $N^\Omega$ is the number of neurons in the output layer, $t_{j,p}$ is the target value of data point $p$ and $o_{j,p}$ is the output of the MLP for data point $p$. In the study described here, we use a MLP with tangens hyperbolicus as activation functions in the hidden layer(s) (here one or two) and linear functions in the output layer trained by the Broy-

den–Fletcher–Goldfarb–Shanno (BFGS) quasi-Newton backpropagation algorithm (Broyden, 1970; Fletcher, 1970; Goldfarb, 1970; Shanno, 1970).

Neural networks are very flexible in terms of number of layers, number of nodes, error metrics, training method, activation function etc.; thus, a series of sensitivity runs was performed, which always consisted of a training and a testing (validation) phase. To find the best network architecture, we varied the following parameters:

– The number and type of input variables (see 2.5)

– The number of hidden layers (one or two)

– The total number of nodes in the hidden layer(s) (between 1 and 14)

To reduce the number of sensitivity runs, the parameters listed in Table 1 were kept fixed based on recommendations in the literature (Zhang, 2008; Kruse et al., 2016).

**2.3   Cross-validation**

Trained networks were validated using $k$-fold cross-validation (Kohavi, 1995; Andersen and Martinez, 1999) to prevent overfitting (Domingos, 2012). Overfitting originates from the trade-off to minimise the error on given data and to maximise performance on new unknown data (Chicco, 2017). To test the network's ability to generalise, the full data set is divided into $k = 6$ subsets by a random data split with approximately equal size first. Cyclically, one subset is kept for independent testing, the
remaining $k - 1$ subsets are used for training and validation. With this test, we can show that ANNs are able to learn from the data and to represent their characteristics. Later on, we will go one step further and check if the found ANNs are able to transfer their knowledge not only on unknown data but on completely new stations not used previously. We decided to validate trained models with the station NL-Cab and to test the best ANNs finally on the station DE-Keh (see details on stations in Chapter 2.4). For these stations, the traditional MOST method performed best; thus, they presented a strong challenge for the
ANNs to achieve similar quality .

**2.4   Data**

To train and validate the neural network, data from 20 observation sites in Europe, Brazil and Russia with different land use types including forest, grassland and crop fields were collected. All data were measured after 2000 and observation periods range from a few months to several years. Figure 1 shows a map of the European sites which provided data. Stations varied
widely in environmental surrounding, instrumental set-up and measurement heights. The tower configuration of the sites is shown schematically in Figure 2.



As ANN input, we used temperatures and wind speed in two measurement heights, and for validation the momentum and sensible heat fluxes to calculate the scaling quantities $u_*$ and $\theta_*$. If not available, density was calculated from the ideal gas equation using virtual temperature in case of available humidity data, otherwise we used the temperature of dry air. For forests, all observations had to be above the canopy. The original temporal resolution of the data was either 10 minutes or 30 minutes; these were aggregated to 1-hour averages.

An important step before using data as input for the ANN was to check if the data were compatible with Monin-Obuhkov theory, i.e. if an (at least approximate) functional relationship between $\zeta$ and the right-hand sides of (1) was to be seen and if yes, how well they were represented by the universal stability functions (1). It turned out that for some sites, no relationship existed. Reasons for this could be a violation of the assumptions of the Monin-Obuhkov theory like inhomogeneous terrain around the site or wind direction dependence of the roughness length. Data from these sites were not used further. The remaining stations (see Table 2) with about 390000 data points in total were used to train and validate the networks. For these, agreement generally was better for temperature than for wind; also, agreement was better for unstable than for stable stratification, which is often mentioned in the literature.

Data were preprocessed before they were presented to the ANN as follows. Input and output data were normalised according to their extrema to the interval $[0, 1]$ (see Table 1). Furthermore, weak wind situations with wind speeds below $0.3\,\mathrm{ms^{-1}}$ were filtered out. Because of large scatter of wind and temperature gradients under atmospheric conditions with absolute heat fluxes below $10\,\mathrm{Wm^{-2}}$ or small scaling wind speeds ($u_* < 0.1\,\mathrm{ms^{-1}}$), such data were excluded. Finally, the signs of the temperature scale $\theta_*$ and of the potential temperature gradient had to be the same, thus excluding counter-gradient fluxes which can be observed over forest (Denmead and Bradley, 1985) and ice (Sodemann and Foken, 2005), but violate the assumptions of MOST (Foken, 2017a, b).

## 2.5 ANN setup and selection of best ANN

We tested network architectures with six and seven element input vectors. The six element input vector consisted of the wind velocity and potential temperature averages over the two heights, the vertical gradients of wind and potential temperature and their ratio and a classifier to distinguish between low ($c_{veg} = 0$) and tall ($c_{veg} = 1$) vegetation. For the seven element input vector, we replaced the temperature gradient by its absolute value and added an additional input node describing the sign of potential temperature gradient. The target vector remained in both cases a two element vector consisting of the wind scale $u_*$ and the temperature scale $\theta_*$.

We experimented with ANNs having one and two hidden layers. For the ANNs with one hidden layer, we varied the number of neurons in the hidden layer from one to twice the size of the input layer. For ANNs with two hidden layers, the number of neurons in each layer is increased up to the number of input neurons.

As mentioned earlier, all networks were trained to minimise the overall (sum of $u_*$ and $\theta_*$) MSE on normalised data from (3). To compare the different ANNs, we used: the root mean squared error (RMSE) $RMSE = \sqrt{MSE}$,



the mean absolute error (MAE)

$$30 \quad MAE = \frac{1}{|P|} \sum_{p \in P} \frac{1}{N^{\Omega}} \sum_{j=1}^{N^{\Omega}} |t_{j,p} - y_{j,p}|$$

and Pearson's correlation coefficient $r$

$$r = \frac{1}{N^{\Omega}} \sum_{j=1}^{N^{\Omega}} \frac{\sum_p (y_{j,p} - \bar{y}_j)(t_{j,p} - \bar{t}_k)}{\sqrt{\sum_p (y_{j,p} - \bar{y}_j)^2} \cdot \sqrt{\sum_p (t_{j,p} - \bar{t}_j)^2}} \quad \in [-1, 1],$$

where $\bar{y}_j$ and $\bar{t}_j$ are the averages of the $j$th net output and the target value with $\bar{y}_j = \frac{1}{|P|} \sum_p y_{j,p}$ and $\bar{t}_j = \frac{1}{|P|} \sum_p t_{j,p}$.

When ANNs are to be used in climate models, one has to find a trade-off between two aspects: on the one hand, the model

should perform well according to the quality metrics described above; on the other hand, a superior model in terms of small errors but with higher computational demands may not be the best choice to use in climate models where saving computing time is a very high priority criterion. For ANNs, computing time normally increases with complexity of a network, i.e. with its size. We therefore tested also ANNs with smaller-than-optimal numbers of neurons in view of this trade-off. To find smaller networks requiring possibly less computing time, we looked at networks that meet the requirement that the size of each hidden

layer $n_{h_i}$ is less or equal to the size of the input layer $n_I$ minus 1.

$$n_I - 1 \ge n_{h_1} \left( \ge n_{h_2} \right) \tag{4}$$

This condition was found after some experimenting and is somewhat arbitrary, but there is no hard rule defining the simplicity of a model. Here, we call ANNs that satisfy this condition simple networks.

## 3  Results

As described in sec. Chapter 2.3, ANNs are always trained on the training data set only and validated on a disjoint validation data set. If the MSE on the validation set rises continuously, training is stopped to prevent overfitting (early stopping). After this training and validation stage, the ability of the thus found ANNs to generalise is tested on data completely new to the ANNs. All in all, more than 100000 nets were trained and tested this way.

### 3.1  Effect of data splitting

The validation results from ANNs with six inputs and one single hidden layer trained under six-fold cross-validation with random data splitting are shown in the box-and-whiskers plot in Figure 3 as a function of of the number of hidden neurons. One can see that the validation MSE decreases with increasing number of hidden neurons and reaches an asymptotic value of about $0.008$ already with 6 to 7 neurons. Furthermore, the scatter of MSE is quite small, meaning that results of ANNs trained on different sets vary only slightly.

If the training data are not split randomly, but station-wise, a larger MSE and a considerably larger scatter of MSE results. Comparing Figure 4 with Figure 3 shows that MSE is roughly doubling, whereas scatter increases by about a factor of ten,



almost independent of the network architecture. On the other hand, increasing the network size doesn't necessarily imply smaller error minima. Using two hidden layers reduces slightly the median and error minimum, but increases the MSE spread, too. Comparison of Figure 3 with Figure 4 also shows that the station-wise error minima are comparable to the ones obtained from random data split. In both types of validation, ANNs with one and two hidden layers are not significantly different. These results show that using the station-wise split reduces network performance substantially. This implies that using not enough stations and station-wise training impairs the portability of learned relationships between inputs and target values. Among the reasons for this could be the tendency of the ANNs to overfitting training data by memorising relationships and local effects

contaminating the validity of MOST like not ideal positioning of sites or not ideal atmospheric conditions. These findings support the need for independent testing with data yet unknown to the ANN in order to estimate the ANNs real ability to transfer knowledge. This will be discussed in the next section.

### 3.2    Transferability to unknown data

After having shown that ANNs are able to extract $u_*$ and $\theta_*$ from training data successfully, our next step is to assess how the

ANNs found in the previous section can handle input from stations which weren't used neither for training nor for validation, i.e. data completely unknown to them; this simulates the situations where ANNs are used in climate models (where grid points play the role of stations). To test this, we choose the station NL-Cab for validation and DE-Keh as the unknown station. These two stations were selected because the standard MOST method performed best for these stations and are therefore a strong challenge for the ANNs to produce equivalent results. The results of the nets performing best on the validation set

are summarised in Table 3, where the ANNs are compared according to increasing complexity of their net architecture. For comparison and in view of reducing computation time, we show in this table also the results of the best simple networks (as defined in section Chapter 2.5). Table 3 shows that in terms of MSE and correlation coefficient $r$, all ANNs perform better than the traditional MOST method on the validation data set (NL-Cab). Applying these ANNs to the test data set DE-Keh results in an increased MSE and lower correlation coefficient, whereas the traditional MOST method performs better on the test data

set. Best test performance of the ANNS in terms of MSE is reached by the 6-5-3-2 ANN with $0.68 \cdot 10^{-2}$, but the simpler 6-3-2 ANN is second best in terms of MSE; it is interesting to see that simple nets can be almost as good as larger nets. ANNs with two hidden layers perform slightly better on the test data than ANNs with a single hidden layer. The overall correlation between network outputs and target values is in all cases quite high ($r \geq 0.85$).

A final comparison is done for the turbulent momentum and heat fluxes $\tau = \rho u_*^2$ and $H = -\rho c_p u_* \theta_*$, which are the quantities

ultimately needed in climate simulations. Results for the momentum and heat fluxes of three well-performing networks as well as for the standard MOST method are shown in Figure 5 and Figure 6 and in Table 4 and Table 5 respectively. Both ANNs and the standard method tend to underestimate larger momentum fluxes, but differences among ANNs are quite small. Best agreement is achieved with the 6-5-3-2 ANN which is almost as good as the standard method; best ANNs and traditional method are quite similar regarding momentum flux.

Also for heat flux, the differences between the ANNs are relatively small, but the ANNs as well as the standard method tend to overestimate the heat fluxes. Best results are obtained with the 6-3-2-ANN. For heat flux, the 7-5-2-2 ANN behaves markedly





different than the other ANNs. It works more like a discrete classifier of stability rather than as a continuous regression: there are two states, one around -30 $\mathrm{Wm}^{-2}$ and the other from 50 $\mathrm{Wm}^{-2}$ to 200 $\mathrm{Wm}^{-2}$. As a result, $r$ is reduced but MAE is lowest for this 7-5-2-2 ANN. These results show again that smaller nets can be as good or even better than larger ones.

A comparison of the computation time required by the different ANNs relative to the 6-3-2 ANN is shown in Table 6. The
table shows that the increase of computational demand is approximately proportional to the number of weights (as could be expected), and therefore increases considerably when two layer networks are used. As the discussion above shows, these costs are not reflected in a markedly higher quality of results.

## 4   Implementation of an ANN in a land surface model

As already mentioned, our goal is to replace the standard MOST method to calculate fluxes by an ANN in the land surface
component of climate models, expecting more flexibility, accuracy and possibly saving of CPU time. The results presented in the previous section indicate that from the accuracy as well as computational efficiency point of view, the 6-3-2 ANN seems to be most suitable for implementation into a land surface model (LSM). This ANN is shown in Figure 7.

The 6-3-2 ANN with weights as found in the previous sections was implemented in a stand-alone version of the one-dimensional LSM Veg3d (Braun and Schädler, 2005) where it replaced the implemented routine using the standard MOST
method to calculate the scaling quantities $u_*$ and $\theta_*$; we will call this version the reference version. Input data for the ANN and data normalisation was the same as described in Chapter 2.4 and output was analogously de-normalised. Since the LSM requires, apart from momentum and heat fluxes, also the moisture flux, the scaling specific humidity $q_*$ was calculated as proportional to $\theta_*$, following the standard procedure in boundary layer meteorology (Arya, 2001). Meteorological input for the LSM was 30" values of short- and long-wave radiation, wind speed, temperature, specific humidity and air pressure at
two heights; additionally soil type and land use were prescribed. For comparison with observations, time series of heat and moisture fluxes as well as soil temperature and soil moisture in the upper soil layers were available. Due to the availability of soil temperature and soil moisture in the top soil layer, the effect of the ANN on the soil component could also be assessed. The comparison was performed with data from the DE-Fal (grassland, year 1988) and DE-Tha (evergreen needleleaf forest, year 2011) stations for years which had not yet been used neither for training nor for validation. We compared the RMSE
and the correlation coefficient of the calculated values with the observed ones for the reference version and the ANN version. Additionally, we compared the required CPU times. The results of the comparison are shown in Table 7 and 8.

Especially for grassland, results of the standard version are very good in terms of RMSE and correlation coefficients and it is difficult for the ANN version to outperform this. However, the results show that the ANN version is able to produce results of similar quality as the standard version for the fluxes as well as for soil temperature and soil moisture. For tall vegetation,
RMSEs are larger and correlation is less; but the differences between the ANN version and the reference version are even smaller than for grassland and for soil moisture the ANN version even outperforms the standard version. In terms of fluxes, the standard version is generally slightly better. Regarding CPU time, there are only minor differences, although we had expected the ANN version to be faster. However, due to the small prognostic time step used, once initialised, the standard version does





in most cases not need to do more than one iteration to find a solution to the nonlinear equation and to update the scaling quantities, so that the expensive iteration is reduced considerably. In summary, as a result of this first comparison one can say that the ANN version works equally well as the reference version.

## 5 Summary

We have used an ANN (more precisely, a MLP) to obtain the scaling quantities $u_*$ and $\theta_*$ as defined in MOST; these are used in weather and climate models to calculate the turbulent fluxes of heat and momentum in the atmospheric surface layer. To train, validate and test the neural network, a large set of worldwide observations was used, representing tall vegetation (forests) and low vegetation (grassland, agricultural terrain). A quality assessment of the data sets showed that not all of them were compatible with MOST, so only 7 of the initially 20 data sets could be used.

Sensitivity studies were performed with different sets of input parameters, training methods and network architectures; validation was done with 6-fold cross validation. An important part of the overall network validation was to check the ability of the network to generalise, i.e. to produce acceptable output if input is data from stations completely unknown to the network. These studies showed that even a relatively small 6-3-2 network with six input parameters and one hidden layer yields satisfying results in terms of RMSE and correlation coefficient. In the trade-off of quality of results vs. computational efficiency, this
network performed best.

We could show that results of the ANN were equivalent to the standard method in all tests we performed. A final validation with the heat and momentum fluxes instead of the scaling quantities showed that the traditional MOST method and the ANN approach were also in this case almost equal in terms of quality, with the 6-3-2 ANN performing best. An implementation of the 6-3-2 ANN into an existing LSM showed that the ANN version gives results equivalent to the standard implementation
with sometimes even higher correlations. However, no saving of computation time was found.

In summary, it could be shown that even in this stage, an ANN gives results comparable in quality to the standard MOST method. Some obvious improvements will include more and better differentiated land use classes (e.g. water, urban areas) and including more situations of strong stratification. Next steps will include more experiments with the input parameters (e.g. including a time lag) and some fine tuning to improve the computational efficiency (e.g. using different activation functions).
We intend to implement and test the neural network routine in a three-dimensional regional climate model. This will require the ANN to learn some additional land use types like urban areas or water surfaces. If these tests are positive, this would open the possibility to replace other "uncertain" components of climate models (e.g. cloud microphysics, sea ice) by neural network subroutines. The main hindrance to do that is presently the lack of suitable training and validation data. An alternative to "real" data could be to use data from more detailed models like LES or urban climate models.

*Code availability.* A MATLAB script (run.m) running the 6-3-2-net with a sample dataset (DE-KaN.dat) can be found under http://doi.org/ 10.23728/b2share.36ef510c515c4a00bb963113647e44a9.



*Data availability.* The data for this study have been obtained from the sources mentioned in the acknowledgements.

5 **Appendix A**

*Author contributions.* LL was responsible for data collection, quality checks and data preprocessing. Also, he trained, validated and generalised the ANNs and compared them with the traditional MOST method. GS implemented the 6-3-2 network in a land surface model and carried out performance measurements in terms of result quality and computational time. GS prepared the manuscript with contributions from LL.

10 *Competing interests.* The authors declare that they have no conflict of interest.

*Acknowledgements.* The authors would like to thank following persons and institutions for providing station data: Martin Kohler and Rainer Steinbrecher (Karlsruhe Institute for Technology), Mathias Göckede, Olaf Kolle and Fanny Kittler (Max Planck Institute for Biogeochemistry Jena), Frank Beyrich (German Meteorological Office), Ingo Lange (University of Hamburg), Clemens Drüe (University of Trier), Marius Schmidt (Forschungszentrum Jülich) and Thomas Grünwald (TU Dresden). In addition, data from following sources were collected and 5 used: Integrated Carbon Observation System Sweden (ICOS), Cabauw Experimental Site for Atmospheric Research (CESAR) database, Oak Ridge National Laboratory Distributed Active Archive Center (ORNL DAAC) and University Corporation for Atmospheric Research (UCAR).





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





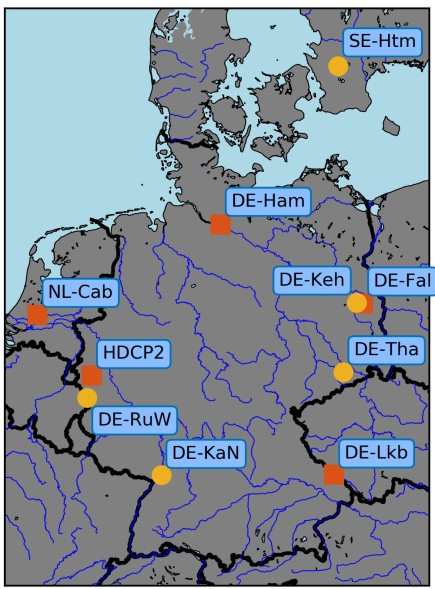

**Figure 1.** Geographical overview on collected station data in Germany and surrounding countries. Station symbols are according to low (red square; grasslands, croplands, wetlands) and tall (yellow circle; forest) vegetation. Stations DE-Nie07, DE-Nie13 and DE-Was06 are combined to HDCP2. Additional stations and further information can be found in Table A1.

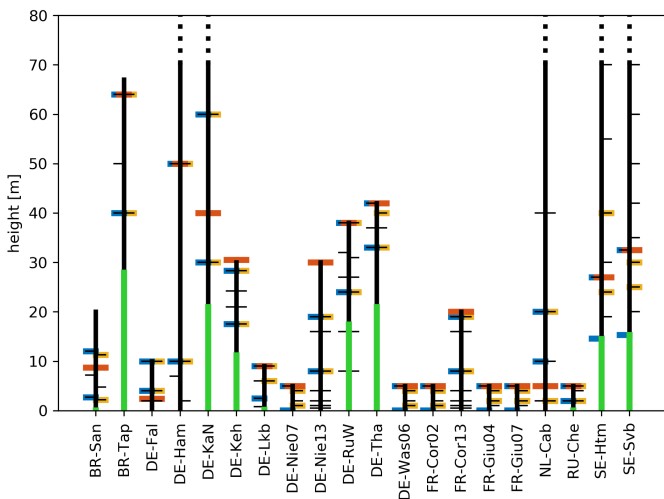

**Figure 2.** Schematic setup of the meteorological towers used for this study. Available measurements for wind velocity (black, left side arm) and temperature (black, right side arm) are shown as well as the finally used measurement height for wind (blue), temperature (yellow) and turbulent fluxes (red). Vegetation height is illustrated in green and towers with a total height above 80m are clipped.





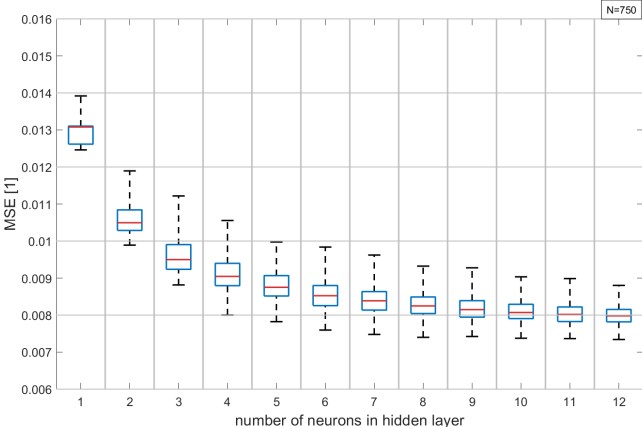

**Figure 3.** MSE of trained networks on validation data set using random data split as a function of hidden layer size. Whiskers indicate interquartile range. Each box summarises results from 750 single networks.

**Table 1.** Fixed network parameters for training.

| | |
|---|---|
| normalisation | $\widetilde{x}_i = (x_i - \min_{Data}(x_i))/(\max_{Data}(x_i) - \min_{Data}(x_i))$ |
| activation function | tangens hyperbolicus |
| activation function output | linear |
| training algorithm | BFGS quasi-Newton backpropagation |
| error metric | MSE |
| early stopping after ... iterations | 50 |
| maximum number of training steps | 1000 |

**Table 2.** Station information of the meteorological towers used Land usage classification follows the International Geosphere-Biosphere Programme (IGBP) standards: evergreen needleleaf forests (ENF), grasslands (GRA), permanent wetlands (WET) and croplands (CRO).

| station | complete station name | lat [°] | lon [°] | height m a.s.l. | IGBP | tower height [m] |
|---|---|---|---|---|---|---|
| BR-San | Santarem Pasture Tower Site (Para, Brazil) | -3.02 | -54.89 | 100 | GRA/CRO | 20 |
| DE-Fal | Grenzschichtmessfeld Falkenberg | 52.17 | 14.12 | 73 | GRA | 10 |
| DE-KaN | KIT CN Messmast | 49.09 | 8.43 | 110 | ENF | 200 |
| DE-Keh | Messstation Forst Kehrigk | 52.18 | 13.95 | 49 | ENF | 30 |
| NL-Cab | CESAR observatory | 51.97 | 4.93 | -0.7 | GRA | 213 |
| RU-Che | Cherksii Tower | 68.61 | 161.34 | 6 | WET | 5 |
| SE-Svb | Svartberget ICOS Sweden | 64.25 | 19.77 | 270 | ENF | 150 |





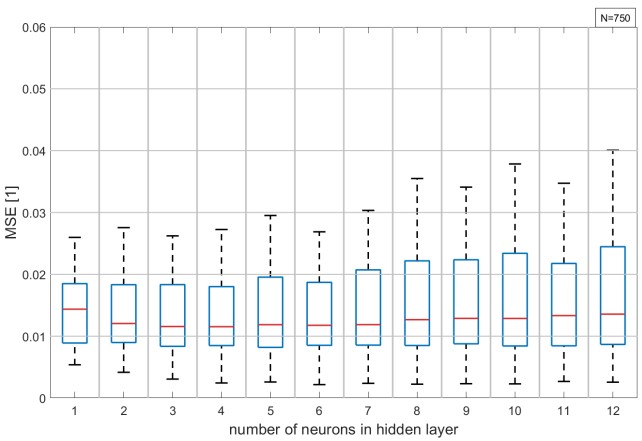

(a) six inputs and one hidden layer

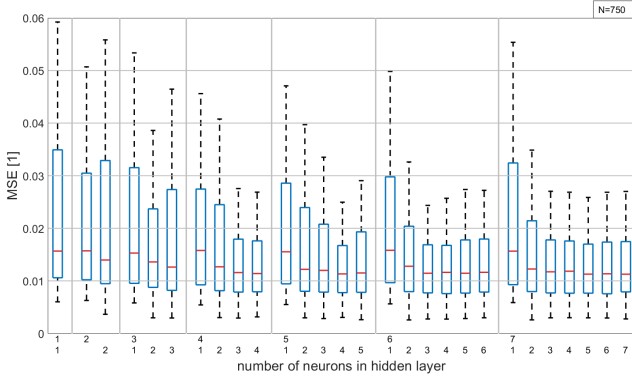

(b) seven inputs and two hidden layers

**Figure 4.** Validation MSE of trained networks using station-wise data split as a function of hidden layer size for (a) the network with six inputs and one hidden layer, (b) the network with seven inputs and two hidden layers. Values for the other networks considered are similar. Whiskers have the length of interquartile range and each box summarises results from 750 single networks each.



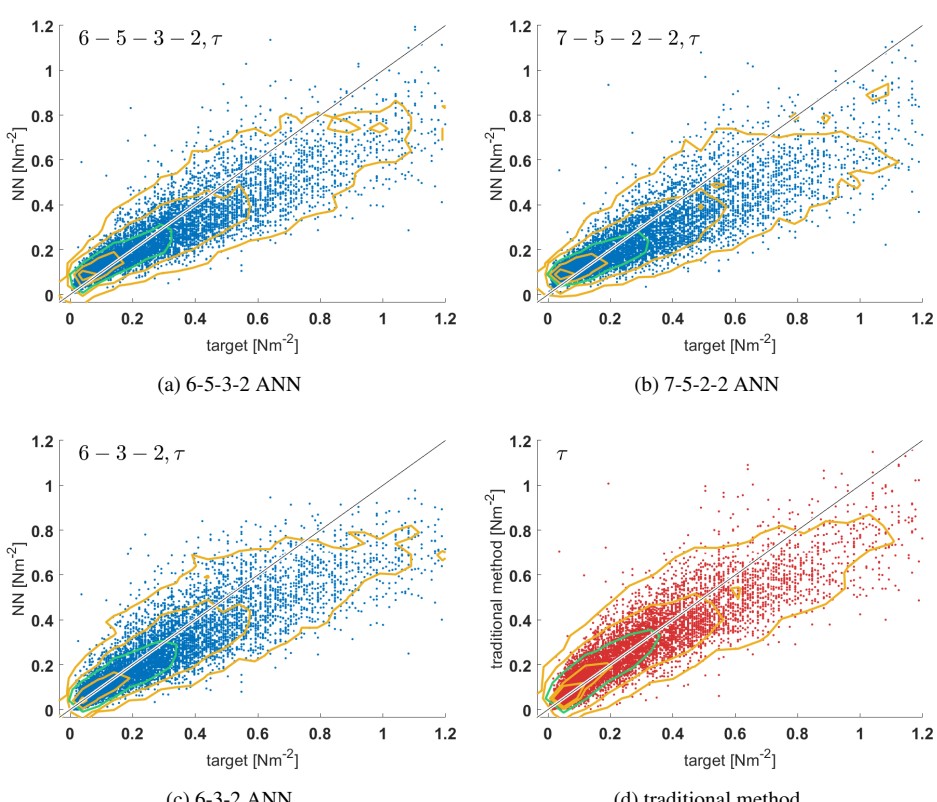

(a) 6-5-3-2 ANN

(b) 7-5-2-2 ANN

(c) 6-3-2 ANN

(d) traditional method

**Figure 5.** Plots of network output versus target values for momentum flux on unknown test data (DE-Keh). Contoured are kernel density estimates of two-dimensional probability density distribution with the 95th, 75th, 25th and 5th percentiles (yellow line) starting outside and the 50th percentile (green).





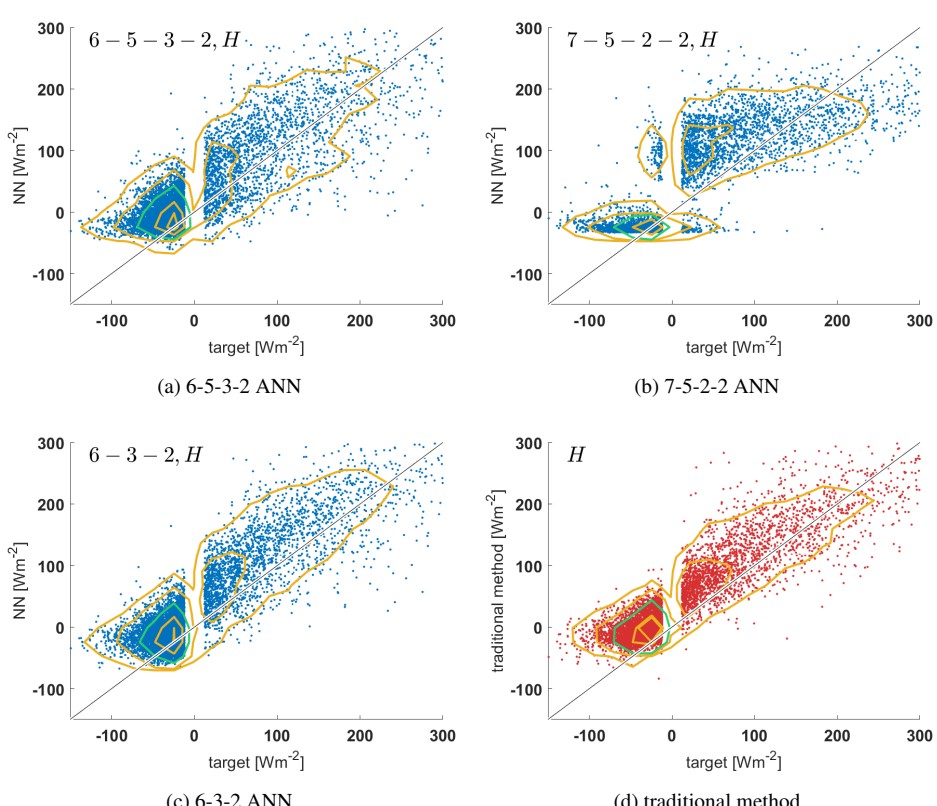

**Figure 6.** Plots of network output versus target values for heat flux on unknown test data (DE-Keh). Contoured are kernel density estimates of two-dimensional probability density distribution with the 95th, 75th, 25th and 5th percentiles (yellow line) starting outside and the 50th percentile (green). The vertical gap is due to the exclusion of heat fluxes between $\pm 10 \mathrm{Wm}^{-2}$.





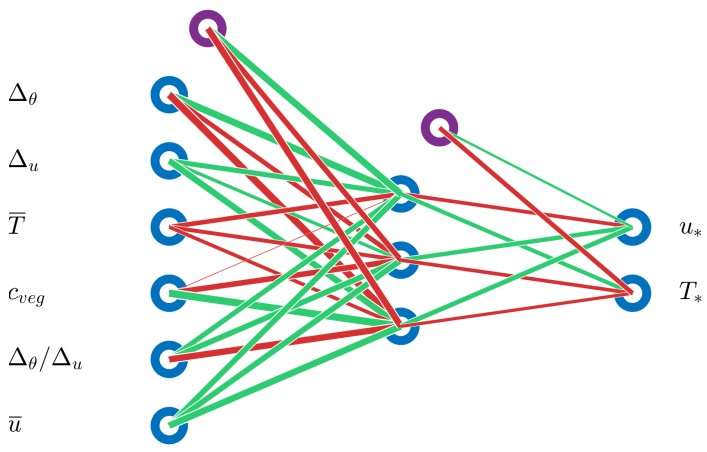

**Figure 7.** The architecture of the 6-3-2 ANN implemented in the land surface model. Input is described in sec. 2.5.

**Table 3.** Performance results of overall best and best simple networks. MSE and $r$ are measured on normalised data and are non-dimensional. $MSE_v$ and $r_v$ are calculated on validation data and $MSE_t$ and $r_t$ on test data. Performance of traditional MOST method (benchmark) is shown too.

| condition | net structure | # weights | $MSE_v[10^{-2}]$ | $r_v$ | $MSE_t[10^{-2}]$ | $r_t$ |
|---|---|---|---|---|---|---|
| overall best net | 6-5-2 | 47 | 0.17 | 0.94 | 0.90 | 0.89 |
| best simple net | 6-3-2 | 29 | 0.38 | 0.92 | 0.74 | 0.87 |
| overall best net | 7-11-2 | 112 | 0.18 | 0.92 | 0.96 | 0.86 |
| best simple net | 7-4-2 | 42 | 0.21 | 0.92 | 1.36 | 0.87 |
| overall best net | 6-5-3-2 | 61 | 0.20 | 0.93 | 0.68 | 0.88 |
| best simple net | 6-3-3-2 | 41 | 0.27 | 0.91 | 0.84 | 0.85 |
| overall best net | 7-5-2-2 | 58 | 0.19 | 0.92 | 0.79 | 0.88 |
| best simple net | 7-4-2-2 | 48 | 0.22 | 0.90 | 1.01 | 0.86 |
| benchmark | - | - | 0.92 | 0.85 | 0.58 | 0.92 |





**Table 4.** Performance of networks vs. standard MOST method (benchmark) for momentum flux.

| net structure | MSE$[10^{-2}\mathrm{N}^2\mathrm{m}^{-4}]$ | RMSE$[\mathrm{Nm}^{-2}]$ | MAE$[\mathrm{Nm}^{-2}]$ | $r$ |
|---|---|---|---|---|
| 6-5-3-2 | 2.11 | 0.15 | 0.09 | 0.90 |
| 7-5-2-2 | 2.44 | 0.16 | 0.10 | 0.89 |
| 6-3-2 | 2.56 | 0.16 | 0.09 | 0.87 |
| benchmark | 1.72 | 0.13 | 0.08 | 0.90 |

**Table 5.** Performance of networks vs. standard MOST method (benchmark) for heat flux.

| net structure | MSE$[\mathrm{W}^2\mathrm{m}^{-4}]$ | RMSE$[\mathrm{Wm}^{-2}]$ | MAE$[\mathrm{Wm}^{-2}]$ | $r$ |
|---|---|---|---|---|
| 6-5-3-2 | 2461 | 49.6 | 37.6 | 0.85 |
| 7-5-2-2 | 2329 | 48.3 | 31.4 | 0.82 |
| 6-3-2 | 2092 | 45.8 | 35.1 | 0.88 |
| benchmark | 1915 | 43.8 | 34.4 | 0.90 |

**Table 6.** Relative computational demand of the ANNs discussed in the text.

| net structure | no. of weights | CPU time (relative to 6-3-2 ANN) |
|---|---|---|
| 6-3-2 | 29 | 1 |
| 6-5-2 | 47 | 1.6 |
| 7-11-2 | 112 | 3.7 |
| 6-5-3-2 | 61 | 2.5 |
| 7-5-2-2 | 58 | 2.4 |





**Table 7.** Comparison of the reference version with the ANN version of Veg3d for the DE-Fal grassland station. $H$ denotes the heat flux, $M$ is moisture flux, $T_s$ is soil temperature, $w_s$ is soil moisture.

|  | Ref | ANN |
|---|---|---|
| CPU time | 10.83 | 10.65 |
| RMSE $H$ [Wm$^{-2}$] | 16.8 | 27.3 |
| $rH$ | 0.87 | 0.81 |
| RMSE $M$ [Wm$^{-2}$] | 15.1 | 20.5 |
| $rM$ | 0.91 | 0.86 |
| RMSE $T_s$ [°C] | 0.8 | 1.3 |
| $rT_s$ | .99 | .99 |
| RMSE $w_s$ [%] | 4.8 | 5.5 |
| $rw_s$ | 0.87 | 0.89 |

**Table 8.** Same as above, but for forest station DE-Tha

|  | Ref | ANN |
|---|---|---|
| CPU time | 95.47 | 97.74 |
| RMSE $H$ [Wm$^{-2}$] | 39.0 | 40.9 |
| $rH$ | 0.52 | 0.57 |
| RMSE $M$ [Wm$^{-2}$] | 27.9 | 33.1 |
| $rM$ | 0.78 | 0.71 |
| RMSE $T_s$ [°C] | 2.4 | 2.2 |
| $rT_s$ | 0.98 | 0.98 |
| RMSE $w_s$ [%] | 5.3 | 3.7 |
| $rw_s$ | 0.53 | 0.75 |





**Table A1.** Station information for all collected meteorological towers. Land use classification follows the International Geosphere-Biosphere Programme (IGBP) standards: evergreen needleleaf forests (ENF), evergreen broadleaf forests (EBF), grasslands (GRA), permanent wetlands (WET) and croplands (CRO).

| station | complete station name | lat [°] | lon [°] | height m a.s.l. | IGBP | tower height [m] |
|---------|----------------------|---------|---------|-----------------|------|------------------|
| BR-San | Santarem Pasture Tower Site (Para, Brazil) | -3.02 | -54.89 | 100 | GRA/CRO | 20 |
| BR-Tap | Tapajos National Forest (Santarem, Para, Brazil) | -3.01 | -54.58 | 100 | EBF | 67 |
| DE-Fal | Grenzschichtmessfeld Falkenberg | 52.17 | 14.12 | 73 | GRA | 10 |
| DE-Ham | Wettermast Hamburg | 53.52 | 10.10 | 0.3 | GRA | 300 |
| DE-KaN | KIT CN Messmast | 49.09 | 8.43 | 110 | ENF | 200 |
| DE-Keh | Messstation Forst Kehrigk | 52.18 | 13.95 | 49 | ENF | 30 |
| DE-Lkb | Lackenberg Messstation | 49.10 | 13.30 | 1300 | GRA | 9 |
| DE-Nie07 | HDCP2 Flux Station 07 Hambach Niederzier | 50.90 | 6.46 | 110 | GRA | 5 |
| DE-Nie13 | HDCP2 Tower 13 Hambach Niederzier | 50.90 | 6.46 | 110 | GRA | 30 |
| DE-RuW | Wüstebach | 50.50 | 6.33 | 621 | ENF | 38 |
| DE-Tha | Anchor Station Tharandt | 50.96 | 13.57 | 380 | ENF | 42 |
| DE-Was06 | HDCP2 Flux Station 06 Wasserwerk | 50.89 | 6.43 | 96 | CRO | 5 |
| FR-Cor02 | HYMEX Flux Station 02 Corte | 43.30 | 9.17 | 369 | GRA | 5 |
| FR-Cor13 | HYMEX Tower 13 Corte | 43.30 | 9.17 | 369 | GRA | 20 |
| FR-Giu04 | HYMEX Flux Station 04 San-Giuliano | 42.27 | 9.52 | 39 | GRA | 5 |
| FR-Giu07 | HYMEX Flux Station 07 San-Giuliano | 42.27 | 9.52 | 39 | GRA | 5 |
| NL-Cab | CESAR observatory | 51.97 | 4.93 | -0.7 | GRA | 213 |
| RU-Che | Cherksii Tower | 68.61 | 161.34 | 6 | WET | 5 |
| SE-Htm | Hyltemossa ICOS Sweden | 56.10 | 13.42 | 115 | ENF | 150 |
| SE-Svb | Svartberget ICOS Sweden | 64.25 | 19.77 | 270 | ENF | 150 |