# Peer review of "Calculating the turbulent fluxes in the atmospheric surface layer with neural networks"

_Geoscientific Model Development, 2018_

## Referee Comment (RC1) · Anonymous Referee #1 · 4 Jan 2019

In my view this is overall an interesting manuscript that introduces a promising novel approach based on neural networks for calculating turbulent fluxes with MOST. With this novel approach, the authors aim amongst others to reduce the computational effort involved in current applications of MOST by circumventing the need for iteratively solving non-linear equations at each time step. Consequently, I believe that many weather and climate models could benefit from the proposed approach in this manuscript because MOST is often applied at the lowest vertical levels to calculate the turbulent fluxes. It is therefore promising that the authors showed that ANNs were able to produce results comparable to the standard method based on MOST.

[Figure]

It is important to note though that an actual decrease in computational time has not been demonstrated yet in this manuscript (as the authors acknowledged), because it required an additional extensive analysis of the proposed approach in at full 3d weather/climate model. Since to my knowledge this is one of the first attempts to apply ANNs to MOST, in my opinion the analysis and results currently presented in the manuscript are nonetheless sufficient for publication. The authors may consider to submit later on a second publication including this additional analysis.

I do have additional specific comments and suggestions regarding amongst others the methodology, citations and clarity of the text that I think should be addressed before publication (especially the comments related to Table 1 and the data used for training). First I will list my specific comments including the line and page number (note that the line numbers were erroneous in the submitted manuscript, and therefore I included the page numbers). After this, I give some grammatical/typographical errors I came across.

Specific comments and suggestions:

Page 1, line 3-7: the first part of this motivation for the ANN is in my view not convincing. The NNs rely on the same small amount of field experiments for training and consequently vary as well, just as the standard empirical MOST equations. NNs trained on 2 different sets of field experiments are unlikely to be the same, to a large extend because of the large uncertainty in the observational data. In short, I think this is more a problem related to the available data rather than the standard MOST method. Also, 'avoiding explicit formulas' can be considered to be actually a disadvantage since it makes it harder to interpret the outcomes from the NN.

Page 2, line 1: see comment above, NNs will likely vary as well depending on the training data.

Page 2, line 7: I think this goal is still a bit too vague. What do you precisely want to improve about the 'overall model performance'? Reducing the computational effort and potentially increasing the accuracy?

Page 2, line 10: I suppose this is observational data, which is not immediately apparent from the text.

Page 2, line 11-12: ANNs are actually non-linear regression models, so I think this statement needs to be rephrased. Furthermore, with 'lagged data' I suppose you mean 'time lagged data'.

Page 2, line 14-15: I think that using different architectures for different years is not feasible in practice as you seem to suggest here, because you do not know for years not present in the training set what architecture will work best. Which architecture performed best overall?

Page 2, line 16-17: See comment earlier, ANN is actually a non-linear regression model. To which (non-)linear regression models did they compare it?

Page 2, line 22-25: This seems to imply that the reduction of inherent variance by the ANN is an advantage. However, Gentine et all. call this an 'unintended side effect' and thus consider this actually to be a disadvantage. Furthermore, it is not clear here what is meant with 'inherent variance': inherent to what? Additionally, Gentine et all did not use one submodel, but rather thousands of cloud resolving models embedded within a climate model. Please rephrase these lines accordingly.

Page 2, line 8-25: I think it is worth mentioning some additional papers here that experiment with the application of ANNs in RANS/LES. LES relies often on MOST at the surface and therefore could be a potential application of your method. Furthermore, the analysis presented in these papers are to some extent in line with the suggestions you make at the end of the conclusions section (i.e using high-resolution model data like LES for training the NNs, which in this case inform lower-resolution models). Examples:

1. Ling, J., Kurzawski, A., & Templeton, J. (2016). Reynolds averaged turbulence modelling using deep neural networks with embedded invariance. Journal of Fluid

Mechanics, 807, 155-166.

2. Vollant, A., Balarac, G., & Corre, C. (2017). Subgrid-scale scalar flux modelling based on optimal estimation theory and machine-learning procedures. Journal of Turbulence, 18(9), 854-878.

3. Sarghini, F., De Felice, G., & Santini, S. (2003). Neural networks based subgrid scale modeling in large eddy simulations. Computers & fluids, 32(1), 97-108.

Page 3, line 8: Replace 'quantity' with 'stability parameter' to make it consistent with the terminology used in chapter 1. Furthermore, $\theta$ should be $\theta$v.

Page 3, line 15: I suggest to clarify here immediately that these stability functions can be used to provide u* and $\theta$* once the stability functions are known. This will connect it better to the previous paragraph.

Page 4, line 1-7: In these lines, you motivate again your approach. Similar to my comments before, I am not convinced that using NNs can reduce the uncertainty: it arises in my view mainly from the large scatter in the data. You can use of course, as you mention here, newer and larger datasets to reduce that uncertainty. However, I don't see the need for ANNs solely for this purpose: the standard method can also 'learn' from newer and larger datasets. I do agree that NNs are a much more flexible method than the standard one, requiring less assumptions about the functional form of the relationship. I see this as an important advantage of NNs, which you may also mention more explicitly in the abstract and introduction.

Page 4, line 12-13: This sentence is not fluent, please rephrase it. Also, you don't mention that the neurons are located in different layers (input layer, one or more hidden layers, output layer), which makes the text sometimes hard to follow. I think a visualization as in Figure 7 would be of great help here for the reader.

Page 4, line 14 equation: You forgot here to include the bias. You can either introduce a separate bias term in the equation, or explicitly specify in the text below that the

summation over the neurons includes the bias neurons. Also, I think you should remove the subscript k: I don't see the added value. Note also that the equation number is missing.

Page 4, line 15: Consider here to replace 'inputs' with something like 'number of neurons in the preceding layer (which includes the bias neurons)'. If you have more than one hidden layer, the input is the preceding layer, not the input layer as 'inputs' implies.

Page 4, line 20 'consistency with the activation functions': In what way? Do you mean that the inputs have an appropriate scale (0-1) for the activation functions?

Page 4, line 25 equation: remove superscript $\Omega$, it seems to me that it has no specific meaning. If it has, please explain it.

Page 5, line 1-9: You could consider to move this to section 2.5, since you discuss it there in more detail. In this section (which focuses on explaining the NN), to me it seems to be out of place. Furthermore, be aware of the distinction between validation and testing (line 2). Rather, you have a training/validation phase and a separate testing phase. Later on, you do make this distinction correctly.

Table 1: Here you list a couple of parameters that are fixed during training based on recommendations from 2 books (please include those references also in the caption). Of those, the early stopping criterion is fixed at 50 training iterations (or do you actually mean 50 epochs?), meaning to me that the training is always stopped after 50 epochs regardless of the complexity of the network. Although this can help to prevent overfitting, it may also cause the training to stop too early, especially for the more complex networks that take much longer to train (and thus need more epochs). A common other approach is that early stopping is done automatically based on a certain stopping criterion which takes into account the training error and validation error. I suggest that you either implement such a 'automatic' early stopping, or that you show some additional plots with the training/validation error during training. The latter should make clear whether training is indeed stopped too early for the more complex networks or

not. Furthermore, I don't completely get what you mean with 'maximum number of training steps'. Does this mean that you stop training after 1000 steps/iterations, in a similar way as the already implemented early stopping criterion? If so, why did you include it besides the already implemented stopping criterion?

Page 5, line 18-20: Are NL-Cab and DE-Keh actually left out of the data used for the cross-validation? I am asking because you mention in line 17 that you wanted to test the ability of the NN to generalize to new stations not seen during training. DE-Keh is however mentioned in figure 1,2 and Table 2, which implies that it is part of the training data used. If De-Keh is in fact used during the cross-validation with random split (and thus seen during training), you cannot claim that you test the ability of the NN to generalize to new stations as done in Section 3.2. Or do you retrain the ANNs presented in Section 3.2 with a training set not including DE-Keh, without cross-validation? I think this needs to be clarified because it can otherwise undermine the analysis presented in Section 3.2.

Table 2: Indicate in the header that these are the stations used for training and validation (and hence not all of them) because only those were compatible with MOST, referring to Section 3.4 for further explanation and Table A1 for all stations. This was not clear to me at first.

Table 2 and A1: indicate also the observation period, such that the reader has a better idea about what data you are using.

Figure 1: Does not include all stations used. You may use separate sub-figures to show all of them. Note that Figure 2 does show all of them.

Page 5, line 21 'observation sites': I suggest that you specify here already that these sites consist of meteorological towers with varying height, such that the reader knows better from the start what kind of data you are using.

Page 5 (where counting from one starts), line 27-28: I suggest that you don't mention
here the input for the ANN, but focus more on the data description itself. It is also redundant because you mention it already in Section 2.5.

Section 2.3 & 2.5: I suggest to put Section 2.5 within 2.3 or vice versa. Both are part of your strategy to select the 'best' ANN, which is now 'interrupted' by the data description. In general, I think the clarity of Chapter 2 will benefit from some reorganization of the content. See also other suggestions.

Chapter 2: What batch size are you using? I cannot find it in this chapter.

Chapter 3, Page 7, line 16: This implies that early stopping is done automatically, while in Table 1 you mention it always stops after 50 iterations (or epochs?).

Figure 3: please indicate in the caption that these results are valid for an ANN with six inputs and one hidden layer under six-fold cross-validation.

Figure 4b: Indicate at bottom axis more clearly that the different numbers represent the two different hidden layers.

Page 7 (where counting from 1 starts), line 27: I think you can, based on Figure 4, make this statement even more general by replacing 'smaller error minima' with 'a lower MSE'. Not only the error minimum, but also for instance the median, doesn't always decrease.

Page 7 (where counting from 1 starts), line 28: I think you can better leave this statement out. When looking at the figures, it is questionable whether this is the case: for higher numbers of neurons, the minima in Figure 4 are around 0.002, while in Figure 3 they are around 0.0075. Also, I think the error minima are less relevant than other features shown in the plots (e.g. the median) since the minima are strongly influenced by outliers and are therefore not representative for the 'common' ANN.

Page 8, line 1 'These results show': In fact, only the observations made in the beginning of the paragraph support this claim. Therefore, I suggest to start here another paragraph and rephrase the sentence in a way like this: 'All in all, the comparison between Figure 3 and 4 shows that the station-wise data split reduces the performance substantially.'

Page 8, line 3: Replace 'portability' with 'generalization', this term is much more common in this context.

Page 8, line 7: Replace 'to transfer knowledge' with 'to generalize', see also comment above

Page 8, line 8: Replace 'transferability' with 'generalization', see comments above.

Table 3: The first column contains a lot of redundant information. I suggest to organize the table differently (grouping 'overall best' and 'best simple' together such you only need to mention it twice)

Page 8, line 9-23: I think it is also worth mentioning that the network with 7 inputs perform worse on the test set than the ones with 6 inputs.

Page 8 line 25-26: This sentence is not fluent and contains too much information. Consider to split this sentence into 2 or more sentences.

Table 4/5: Mention in the caption these results are from DE-Keh

Page 8, line 28-29: Remove second part sentence after ';' : it is redundant because it basically repeats the first part of the sentence.

Page 9, line 2-3: This sentence may suggest to some readers that the 7-5-2-2 ANN is favourable because MAE is lowest, even considering the next sentence. You probably don't intend to write this, so I think it is good to put here more emphasis on the fact that the discrete classifier behaviour is unwanted and thus the 7-5-2-2 ANN is not favourable to use.

Page 9, line 7: This is I think to a large extent caused by the lack of (high-quality) training data, which is worth mentioning here.

[Figure]

Section 3.2: In my view this section lacks a summary/conclusion (that deals with generalization as the header implies) and a coherent message the reader should remember. In Section 4 you argue that the 6-3-2 ANN is best in terms of accuracy and performance, so I think it would be good to mention that here already as well. Note that you did properly include such an intermediate summary in Section 3.1.

Page 9, line 13-15: If I interpret this sentence literally, it says to me that the reference version contains the ANN rather than the standard MOST method. I advise you therefore to rephrase this sentence.

Page 9, line 19 " 30" ": you probably mean here 30 seconds in time, not 30 arcseconds. To avoid any possible confusion, I suggest to simply write " 30s ".

Page 9, line 23: Note that DE-Fal is mentioned to be part of the training data (Table 2). Similar to my comment at Page 5, line 18-20, it is again a bit confusing to me what data is used for training and testing. Try to be more specific and less ambiguous about this. Furthermore, I also note that DE-Tha was not included in the previous analysis because it was not compatible with MOST (Section 2.4). This deserves an additional explanation why you can use this station in this analysis while you couldn't use it in the previous analysis.

Figure 7: With the purple points you probably indicate the bias neurons, please indicate this more explicitly

Page 10, line 10: In your paper I did not see any sensitivity study with respect to the training method. In fact, Table 1 mentions that the used training algorithm was fixed. Therefore, you cannot claim that you did this.

Page 10, line 14-15: Sentence is not very fluent. My suggestion is to say something like: "In view of the trade-off between ... and .... , ..."

Page 10, line 26-29: Despite that a couple of papers already attempted this to some extent (e.g. Gentine et al. 2018 as cited in your manuscript, see previous comment at

Page 2, line 8-25 for more papers), I agree that still a lot of work needs to be done in this regard and therefore your (future) contributions can be very relevant. Maybe you can cite some of these papers here (again) as they can make your suggestions more credible (since it shows that researchers are already attempting it).

Grammatical/typographical errors and suggestions:

Overall: use either 'nonlinear' or 'non-linear', be consistent in present/past tense and active/passive form (at least within individual sections)

Page 1, line 3: remove '(resp Earth system) models', add something like 'climate and weather forecast models'

Page 1, line 18: remove brackets, write something like 'climate and weather forecast models'

Page 1, line 19: replace 'fluxes respective covariances' with 'fluxes'

Page 2, line 2: replace 'resp' with 'and/or'

Page 2, line 5-25: Be consistent with present and past tense.

Page 2, line 20: is –> is given in

Page 2, line 28: Consider to put the parts between brackets as a footnote.

Page 4, line 16: A network becomes non-linear –> In a network non-linearity is included by

Page 4: equation number missing

Page 5, line 17: 2x on –> to

Page 6, line 2: was to be seen –> was present

Page 6, line 9: as follows –> as follows : , or remove 'as follows'.

Figure 3 and 4 label: MSE[1] –> MSE

[Figure]

Figure 4 caption: whiskers have –> whiskers indicate, remove second 'each'

Page 7, line 21: of of –> of

Page 7, line 23: results of ANNs trained ... vary –> the quality of the results from ANNs trained ... varies

Page 7, line 25: remove first ','

Page 8, line 4: overfitting –> overfit

Table 3 caption: too –> Also, ...

Section 3.2: sometimes in activate form, mostly in passive form. Try to be consistent.

Page 8, line 30: Also for heat flux, –> Regarding the heat flux,

Page 9, line 16-18: Remove all ',' except the third one.

Section 4: active and passive form used interchangeably, try to be consistent.

---

## Author Comment (AC1) · 23 Jan 2019

Response to referee #1 23.1 2019

We thank the anonymous referee #1 for the comprehensive and sound comments which certainly helped to improve and clarify the paper. Our responses follow the order of the comments.

A revised version of the paper can be found in the supplement.

p1,3-7: we agree that data availability is a problem for the standard regression as well as the ANN method; however, the data sets we used were considerably larger than the

ones used

previously and were checked for consistency with MOST. The ANN method seems more flexible than standard multivariate regression because no assumptions about the functional form of the

relationship are made. Having explicit formulas instead of ANNs would only be an advantage if there were some physics behind the formulas, which is not (yet) the case.

p2, 1: we agree

p2, 7: rephrased more explicitely

p2, 10: corrected

p2, 11-12: text changed

p2, 14-15: text adapted. We do not suggest anything, we just quote the paper. Overall best model is not mentioned in the paper.

p2, 16-17: text changed

p2, 22.25: text changed

p2, 8-25: text extended

p3,8: we do not agree here: the quantities mentioned are not stability parameters. We used potential temperature, therefore no index v.

p3, 15: done

p4, 1-7: moved to introduction and rephrased

p4, 12-13: rephrased

p4, 14: done

p4, 15: done

p4, 20: yes; rephrased

p4, 25: we used the superscript Omega to make clear that N refers to the output ("Omega") layer

p5, 1-9: done

Table 1: we rephrased and extended the text

p5, 18-20: we rephrased the description. DE-Keh was left out in training and validation in the second experiment

Table 2: done

Table 2 and A1: done

Figure 1: replaced

p5, 21: done

p5, 27-28: done

Sec 2.3&2.5: Moved "Cross-validation" after "Data"

Ch 2: batch is the whole training set

p7, 16: rephrased in sec 2.5.

Fig 3: done

Fig 4b: done

p7, 27: done

p7, 28: which statement do you mean?

p8, 1: rephrased

p8, 3: done

p8, 7: done

p8, 8: done

Table 3: table rearranged

p8, 9-23: this is not genarally true. We added a sentence "Networks with seven inputs have in our case no substantial advantage over networks with six inputs."

p8, 25-26: we don't get what you mean here

Table 4/5: done

p8, 28-29: done

p9, 2-3: rephrased

p9, 7: we do not agree here because: a) the training data set we used was quite large, and b) using even more and even better training data would probably also improve the results of the

simpler nets, so cost/benefit might not change.

Sec 3.2: brief summary added

p9, 13-15: rephrased

p9, 19: input was every 30 min, corrected

p9, 23: data were new in the sense that time periods were used which had not been used for training and validation; the DE-Tha site had not been used at all before, because a) the sites

selected in sec 2.4 were more consistent with MO than DE-Tha and b) the DE-Tha time series covered only one year. For running the LSM for the DE-Tha site, a more comprehensive input data

set (including e.g. radiation, precipitation) was required, which was only available for

the year 1998. The years for the two sites were mixed up in the paper: it should be 2011 for the

DE-Fal site and 1998 for the DE-Tha site. (not 2011, as the paper says erroneously).

Fig 7: done

p10, 10: replaced "training method" with "data sampling method"

p10, 14-15: done

p10, 26-29:

grammar/typos:

overall: we use nonlinear

p1, 3: done

p1, 18: done

p1, 19 : done

p2, 2: done

p2, 5-25: changed to present tense consistently

p2, 20: sentence rephrased

p2, 28: we prefer to leave it as it is

p4, 16: done

p4, eqn#: done

p5, 17: done

p6, 2: done

p6, 9: done

Figs 3 and 4: done

Fig 4, caption: done

p7, 21: done

p7, 23: done

p7, 25: done

p8, 4: done

Table 3: done

Sec 3.2: changed to active form

p8, 30: done

p9, 16-18: done

Sec 4: changed to active form

Please also note the supplement to this comment:
https://www.geosci-model-dev-discuss.net/gmd-2018-263/gmd-2018-263-AC1-
supplement.pdf

**Supplement:**

[revised manuscript text omitted]

---

## Referee Comment (RC2) · Anonymous Referee #2 · 28 Jan 2019

Review of: Calculating the turbulent fluxes in the atmospheric surface layer with neural networks =========================================================================

The paper describes a boundary-condition for specifying turbulence fluxes in climate models, as a function of readily available features. The BC is entirely empirically specified: an ANN is trained from experimentally measured data, and the resulting predictions are compared with those of a standard model (MOST).

The paper is generally well written, and addresses the potentially interesting and important topic of wall-modelling in complex flows, but suffers from a lack of analysis of the data and performance of the numerical method proposed. Coarsely speaking this

work fits the pattern of: (i) apply machine-learning to data-set, (ii) report the fit. Little insight into the data, physics, or performance of the algorithms is gained by the reader - and the authors' own results suggest a much simpler model would fit their data equally well. A path they do not investigate. For these reasons I recommend rejection.

Major comments:

- The most serious shortcoming, is that of more-or-less uncritically applying ANNs to the data-set, without examining their suitability, and to what extent the data can be explained by simpler models. In particular Figure 4 shows performance of your ANNs which is almost independent of number of neurons. In fact you show a 1-layer, 1-neuron "network" performs basically the same as 1-layer, 12-neurons, or a deep network with 2-layers with 7-7 neurons (7 inputs). This result strongly suggests that almost all the predictive power of ANN for this data is contained in a linear fit. At the very most a linear-fit-plus-bounds would have essentially the same predictive power (given your use of the tanh activation function).

Given this, it seems redundant and unnecessarily complicated to use the heavy-machinery of ANNs, with its associated costs: obfuscation of the functional relationship, expense of evaluating the network, and lack of statistical/noise modelling.

Indeed, data that can be reproduced with a single perceptron strongly suggests an extremely simple main relationship between features and output, which is an opportunity to discover simple physical relationships and main-effect parameters.

Note that Figure 3 is not a defense against these criticisms - as the authors themselves state, the relevant plot for the usefulness of the ANN in climate models is Figure 4, not Figure 3. Indeed the difference between these figures indicates that the more complex networks are overfitting the data from the available towers.

In my opinion this paper should not be published without an analysis of the data and a comparison with simpler models. Data analysis could include:

[Figure]

**GMDD**

o Sobol index analysis to identify effects of individual features and coupled features on the output (ANOVA). o Active-subspace analysis to identify main-directions in the input space that contribute most to the output (these will most likely correspond to the weights in your 1-layer, 1-neuron network). o Correlation analysis on the input-space, are inputs independent? This and last two points will contribute to dimensional reduction. o Parametric/Non-parametric estimation of noise in the output.

Simpler models could include:

o Linear regression, with a variety of noise models. o Linear-regression-plus-bounds (only if the above fails) o Input variable elimination (via ANOVA) prior to linear regression/ANN. o Low-order polynomial fits/gene-expression programming to obtain simple explicit expressions capturing the functional relationship.

Only if ANNs do significantly better than linear models is the current work worth publishing.

- I'm not convinced by the assertion that there is a significant computational speed advantage to be gained by replacing MOST with an ANN. I'm not familiar with global climate models, but I see the intent to use it as a BC in an 3d LES simulation. In similar simulations in engineering problems, wall-modelled BCs (e.g. involving solution of an ODE at the wall) account for a negligible component of the total CPU time, and never more than $\sim$5%, with most time spent in the volume. Please explain what is special about your models that causes this situation to be reversed. Please quantify the time spend by your code in various parts of the calculation, so the reader can see the relation of the ground-modelling to other time-consuming parts of the code.

- Assumptions: Repeated reference is made to the assumptions made in the derivation of MOST. I would appreciate in Section 2 an enumeration of all assumptions made, perhaps with some comment on their validity and their role in simplifying the MOST model.

[Figure]

- Relatedly, the training data-sets contain the turbulence fluxes. Are these fluxes measured directly, or are they computed from measurements of u* and theta*, or is a more complex model used to map from measured quantities to turbulent fluxes. What assumptions are made in this map? What modelling assumptions are inherent to your ANN approach?

- Title should be "turbulence fluxes" not "turbulent fluxes". They are fluxes-of-turbulence, not fluxes-which-are-turbulent.

---

## Author Comment (AC2) · 4 Feb 2019

Reply to referee #2: Our reply is structured like this: we quote the essential part of the referee's comment in inverted commas, followed by our reply

"... BC is entirely empirically specified ...": the BC (or rather $u^*$ and $T^*$) is derived on the basis of MOST, and as we state in sec 2.1, on this basis our goal is to determine $u^*$ and $T^*$ from known quantities, which are in our case modelled or observed wind and temperature gradients in the surface layer. So it's not "entirely empirical".

"Little insight into the data ...": data have been described and checked carefully for

compatibility with MOST (sec. 2.3). Underlying physics is MOST, described in sec. 2.1. Performance of the algorithms is discussed in secs. 3 and 4.

"... and the authors' own results suggest a much simpler model would fit their data equally well. A path they do not investigate.": that was not our aim. Our aim in this paper was to see specifically if and how well the stability functions could be approximated by an ANN.

"... is that of more-or-less uncritically applying ANNs to the data-set, without examining their suitability, and to what extent the data can be explained by simpler models": we gave the reasons why we tried ANNs: see remarks above and secs. 1 and 2.1 of the paper. What would a critical application look like in the opinion of the referee?

"In fact you show a 1-layer, 1-neuron "network" performs basically the same as 1-layer, 12-neurons, or a deep network with 2-layers with 7-7 neurons (7 inputs).": this is not the case.Especially in fig. 3 one can see a substantial trend that a network with one single hidden neuron is outperformed by networks with several hidden neurons. Also fig. 4 shows this trend in an attenuated pattern.

"This result strongly suggests that almost all the predictive power of ANN for this data is contained in a linear fit.": a linear fit would certainly not work. Stability functions are highly nonlinear, see formulas in sec. 2 and e.g. Arya's book.

"Given this, it seems redundant and unnecessarily complicated to use the heavymachinery of ANNs, with its associated costs": we don't think ANNs can be considered heavy machinery nowadays; the difficult parts of the work are a) obtaining and filtering data, and b) validation and testing - this has to be done for all kinds of regression methods.

"... an extremely simple main relationship between features and output:" the task is indeed simple: approximate a single valued nondimensional function of one variable (which is a nondimensional combination of other variables). The essential physics is

captured in the Monin-Obukhov length and the dimensionless gradients (i.e. stability functions).

"Indeed the difference between these figures indicates that the more complex networks are overfitting the data from the available towers.": we discuss our use of the (less complex) 6-3-2 ANN in secs. 3 and 4.

"... comparison with simpler models ...": as explained above, this was not our aim; a comparison is done in secs. 3 and 4 with the regression (not physics!) based functions in the literature.

"Only if ANNs do significantly better than linear models is the current work worth publishing.": why?

"I'm not convinced by the assertion that there is a significant computational speed advantage to be gained by replacing MOST with an ANN ...": we are not sure here either - but we would like to try. This was the first step - next step will be implementation in a regional climate model (RCM).

"3d LES simulation": we do not intend to do LES simulations.

"Please explain what is special about your models that causes this situation to be reversed. Please quantify the time spend by your code in various parts of the calculation": The situation is not reversed. Climate models, especially RCMs, are very expensive to run (climatologically relevant multidecadal simulations at high resolution can take several tens of weeks on a high performance system), so every saving is valuable, especially in view of the other advantages. We hope to save around five percent (i.e. about one week), taking into account parallelisation.

"I would appreciate in Section 2 an enumeration of all assumptions made, perhaps with some comment on their validity and their role in simplifying the MOST model.": this is done in the data section: ... Reasons for this could be a violation of the assumptions of the Monin-Obukhov theory like inhomogeneous terrain around the site or wind direction

dependence of the roughness length.

"Are these fluxes measured directly": at all sites used, fluxes are measured by the eddy covariance method, from which u*and T* are derived with the formulas from sec. 2.

"What modelling assumptions are inherent to your ANN approach?": this is explained in sec. 2.1. "Title should be "turbulence fluxes" not "turbulent fluxes".": we would like to stick to the terminology used in the boundary layer meteorology community, which is "turbulent fluxes" (see e.g. Arya's book).

---

## Author Response (AR3)

gmd 2018-263

Reply to the editor

Dear Dr. van Heerwaarden,

thank you for your comment which we quote here: " I would like you to take into account this point, and after satisfactory implementation, I think the paper can be published.

I agree with reviewer #2 that Figure 4 indeed suggests that a very simple model (perhaps even a linear one because of the usage of the tanh activaiton function) would do the trick."

We have added a few sentences regarding this comment in our manuscript (marked in green in the latest version).

Let us add a few more remarks on that issue: our comparison with the multivariate linear regression model shows that it performs considerably worse than all ANNs. We think that this is no surprise, since nonlinear functions have to be fitted which is not well possible with any linear model. Capturing nonlinearity seems to us to be the crucial point. Even our simplest ANN with a small number of hidden nodes, but with the nonlinear tanh activation function, is a nonlinear model and seems to be better suited for the problem mainly for this reason. That already a small ANN gives good results may be due to the "easy complexity" of the problem, but doesn't take away from the nonlinearity.

All "simpler" suggestions of reviewer #2 except the first are also nonlinear models and the wide range of these suggestions indicates a certain arbitratiness, which in our opinion can be avoided to a large extent by using ANNs.

Best regards,

Gerd Schädler and Lukas Leufen